# Alcohol Use and Its Related Psychosocial Effects during the Prolonged COVID-19 Pandemic in Japan: A Cross-Sectional Survey

**DOI:** 10.3390/ijerph182413318

**Published:** 2021-12-17

**Authors:** Nagisa Sugaya, Tetsuya Yamamoto, Naho Suzuki, Chigusa Uchiumi

**Affiliations:** 1Unit of Public Health and Preventive Medicine, School of Medicine, Yokohama City University, Yokohama 236-0004, Japan; nagisa_s@yokohama-cu.ac.jp; 2Graduate School of Technology, Industrial and Social Sciences, Tokushima University, Tokushima 770-8502, Japan; uchiumi@tokushima-u.ac.jp; 3Graduate School of Sciences and Technology for Innovation, Tokushima University, Tokushima 770-8502, Japan; gypll.82@gmail.com

**Keywords:** the coronavirus disease 2019, alcohol use, the Alcohol Use Disorders Identification Test, mental health

## Abstract

We conducted a large-scale survey in the Japanese population, about one year after the initial declaration of the state of emergency, to investigate alcohol use under the prolonged coronavirus disease 2019 (COVID-19) pandemic and its related psychosocial and demographic characteristics. The survey was conducted online between 15 and 20 June 2021. A total of 11,427 participants were included in the analysis (48.5% female, 48.82 ± 13.30 years, range = 20–90 years). Compared with females, males were more prevalent in the hazardous user and the potential alcoholism group and were less prevalent in the no alcohol-related problem group. However, the prevalence of potential alcoholism among the participants in our study was higher than that previously reported. This trend was particularly pronounced in women. The presence of potential alcoholism was related to a deteriorated psychological status, particularly depression and anxiety, and various difficulties in their daily lives due to the COVID-19 pandemic. In the future, intervention methods and systems should be developed to provide optimal assistance to people with psychological problems who are vulnerable to alcohol-related problems during the COVID-19 pandemic, while conducting further long-term follow-up studies.

## 1. Introduction

Coronavirus disease 2019 (COVID-19) has rapidly spread worldwide since its outbreak in December 2019. To deter the spread of COVID-19, many countries have imposed repeated lockdowns with restrictions on outings, service closures, and so forth. While lockdowns are expected to deter the spread of infection, they also cause psychological distress and economic damage [1,2].

Many studies have been conducted on alcohol use during the COVID-19 pandemic and its lockdown in many countries [3,4]. Alcohol use was reported to be one of the factors associated with higher odds of suicidal behavior during the COVID-19 pandemic [5]. Although some studies in Japan, India, the United States, and Czechia reported that alcohol use decreased [6,7] or that alcohol-related problems did not change [8,9] during the COVID-19 pandemic, other studies in the United States and Norway reported that alcohol use or alcohol-related problems increased [10,11,12,13]. Findings from eight European countries indicated that during the first months of the COVID-19 pandemic, alcohol consumption in the upper decile of drinkers increased, as did the prevalence of heavy drinkers, in contrast to the declining consumption in other groups in the sample [14]. Previous research in the United States found that participants reported drinking more alcohol due to having more time (28%) or boredom (22%) [15]. Studies from Italy and Belgium reported that increased alcohol consumption was particularly prevalent in men [16,17]. Among women in Germany, 23% increased their alcohol consumption during the COVID-19 pandemic, and the presence of generalized anxiety was a significant predictor of increases in alcohol use [18]. Another study from Germany reported that consuming more alcohol during lockdown was associated with middle age, higher subjective stress due to the COVID-19 pandemic, a lower agreement with the importance of the restrictions, and consuming alcohol more than once per week before the lockdown [19]. Previous results in Norway showed that people aged 30–39 years had higher odds of increased alcohol consumption during lockdown compared to the oldest adults, and increased alcohol consumption was more frequent among people reporting economic troubles, those quarantined, and those studying or working at home [20]. A previous study estimated that there were 3 million alcohol-attributable deaths in 2016 globally, and reported that alcohol use was a major risk factor for communicable, maternal, perinatal, and nutritional diseases, non-communicable diseases, and injury deaths [21]. Therefore, it is important to understand worldwide problems of alcohol use during the COVID-19 pandemic in order to take measures to protect people’s health.

The Japanese government repeatedly declared a state of emergency in April 2020. While many countries were in lockdown with penalties for violations, the Japanese policy for COVID-19 differed as the government did not impose penalties and only requested people to refrain from going out, except for emergencies, and to temporarily limit certain businesses. However, the “mild lockdown” [22] in Japan affected people’s lives in many ways, causing changes in their lifestyle due to teleworking or online classes, and economic damage due to decreased income or job loss [23]. The prolonged COVID-19 pandemic and repeated declaration of states of emergency may significantly deteriorate the psychological condition of the Japanese population [24]. These long-term stressful conditions and behavioral restrictions may affect alcohol use in the Japanese population. In a recent survey in Japan, the percentage of those who drank alcohol in amounts that increased their risk of lifestyle-related diseases (women: 20 g/day or more of pure alcohol; men: 40 g/day or more) decreased between 2013 and 2018, but did not change for women [25]. A nationwide survey in Japan conducted in 2018 suggested that alcoholism was prevalent among 40,000 women and 220,000 men, of which around 80% did not seek medical care [25].

A previous study [6] on dietary patterns including alcohol use in the Japanese population was conducted during the early stages of the COVID-19 pandemic (January–May 2020), and it reported that their alcohol intake decreased. However, no previous research has investigated problematic alcohol use during the prolonged COVID-19 pandemic and its demographic and psychosocial factors in Japan using large-scale samples, to the best of our knowledge. Moreover, a survey conducted later in the pandemic, such as 1 year after, may yield different results from the previous studies described above. Thus, we conducted a large-scale survey in the Japanese population about one year after the initial declaration of the state of emergency to investigate alcohol use during the prolonged COVID-19 pandemic and its related psychosocial and demographic characteristics. 

## 2. Materials and Methods

### 2.1. Participants and Data Collection

The survey was conducted online between 15–20 June 2021. The survey period was during the third state of the emergency. We conducted an online survey with the following inclusion criteria: (a) inhabitants living in the six prefectures (Tokyo, Aichi, Osaka, Kyoto, Hyogo, and Fukuoka) and (b) age ≥ 20 years. The exclusion criteria were (a) age < 20 years and (b) high school students. The number of participants from each prefecture was determined based on the ratio of the prefecture’s populations—Tokyo (*n* = 3763; 32.9%), Aichi (*n* = 1827; 16.0%), Osaka (*n* = 2360; 20.6%), Kyoto (*n* = 618; 54.0%), Hyogo (*n* = 1516; 13.2%), and Fukuoka (*n* = 1369; 12.0%). 

Through Macromill.inc. (Tokyo, Japan), a global marketing research company, participants were recruited via email, and data were collected on an online platform. Participants completed the online survey after having received a link. All participants voluntarily responded to the survey anonymously and provided their informed consent online before completing the survey. Participants received a clear explanation of the survey procedure and could interrupt or terminate the survey at any time without providing a reason. The questionnaire format, excluding the default items provided by Macromill, Inc. (sex, age, occupation, annual household income, marital status, and presence of children), did not allow participants to proceed to the next page if there were items they had not answered. All participants received Macromill points for their participation, which could be exchanged for prizes or cash. Macromill points are a points-based reward system provided by Macromill Inc. 

This study was approved by the Research Ethics Committee of the Graduate School of Social and Industrial Science and Technology, Tokushima University (acceptance number 212) and was performed per the ethical standards of the 1964 Declaration of Helsinki and its later amendments.

### 2.2. Measurements

#### 2.2.1. Sociodemographic Characteristics

We collected sociodemographic information in participants including age, sex, employment status (employed, homemaker, student, unemployed, or other), marital status, the presence of child, and annual household income (<2.0 million JPY, 2.0–3.9 million JPY, 4.0–5.9 million JPY, 6.0–7.9 million JPY, ≥ 8.0 million JPY, or unknown). 

#### 2.2.2. Alcohol Use

Alcohol use was assessed using the Japanese version of the Alcohol Use Disorders Identification Test (AUDIT) [26]. The AUDIT has 10 items across three domains: hazardous alcohol use (three items), dependence symptoms (three items), and harmful alcohol use (four items). Each item was scored from 0 to 4, 0, 2, or 4. The lowest possible score of the AUDIT was 0, and the highest possible score was 40. The likelihood and severity of hazardous drinking, harmful drinking, and alcohol dependence increased as the score increased. Persons with a score of 8 points or higher in the AUDIT were considered to have a hazardous drinking problem (hazardous user group), and those with a score of 15 points or higher were considered to have alcohol dependence (potential alcoholism group) based on the cut-off criteria of the AUDIT by the World Health Organization and in specific health guidance by the Japanese Ministry of Health, Labour and Welfare [27,28]. Persons with a score of 7 points or lower were defined as those without alcohol problems (no problem group). Previous research conducted in the Japanese population before the COVID-19 pandemic reported that males had higher AUDIT scores than females [29].

#### 2.2.3. Psychological Distress

We used the Japanese version of the Kessler Psychological Distress Scale-6 (K6) [30], which is a six-item screening instrument measuring non-specific psychological distress over the past 30 days. Each question was rated on a scale of 0 (none of the time) to 4 (all of the time), with total scores ranging from 0 to 24. Because of its high accuracy and brevity, the K6 is considered an ideal scale for screening for mental disorders or psychological distress in population-based health surveys [30,31,32]. K6 scores ranging from 5 to 12 were defined as mild-to-moderate psychological distress. This threshold is the optimal lower threshold cutoff point for screening for moderate psychological distress [33]. A threshold score of 13 was traditionally used in previous studies [31,34]. A score of ≥13 was defined as serious psychological distress, and a score of ≤4 was defined as no or low psychological distress. 

#### 2.2.4. Depression Symptoms

We used the Japanese version of the Patient Health Questionnaire-9 (PHQ-9) [35] to evaluate the symptoms of depression amongst the participants; the PHQ-9 comprised nine questions. Symptoms of depression during the past four weeks were reported by the participants, with a score of 0 (not at all) to 3 (nearly every day) [36]. A score of ≥10 was recommended as a cutoff point to indicate that a person was more likely to have major depression. 

#### 2.2.5. Anxiety

We used the Japanese version of the Generalized Anxiety Disorder-7 (GAD-7) [37] to assess anxiety symptoms. Seven questions were rated on a scale of 0 (Never) to 3 (Almost every day). The total score ranged from 0 to 21 points [38]. In the evaluation of symptoms, a score of 0 to 4 points indicates no anxiety disorder, 5 to 9 indicates mild anxiety, 10 to 14 indicates moderate anxiety, and 15 to 21 indicates severe anxiety disorder. A respondent who scored 10 points or more was considered to require drug therapy.

#### 2.2.6. Loneliness

We measured loneliness using the Japanese version of the UCLA Loneliness Scale Version 3 (UCLA-LS3) [39]. The UCLA-LS3 consists of 10 items, and each question was rated on a scale of 1 (never) to 4 (always) [40]. The total scores ranged from 10 to 40, with higher scores indicating higher levels of loneliness. 

#### 2.2.7. Social Isolation

We measured the social networks of the participants during the state of emergency using the Japanese version of the abbreviated Lubben Social Network Scale (LSNS-6) [41]. The LSNS-6 is a shortened version of the Lubben Social Network Scale [42] that includes six items on the network size of relatives or friends who provide emotional and instrumental support. The LSNS-6 consists of three items related to the family and friendship network each. Each question was rated on a scale of 0 (none) to 5 (nine or more) [43]. The total score ranged from 0 to 30 points, with higher scores indicating a larger social network and <12 points indicating social isolation. 

#### 2.2.8. Lifestyle, Coping Behavior, and Stressors Related to the Mild Lockdown

With extensive references to the literature on the COVID-19 pandemic [44,45,46,47,48], we developed eight lifestyle and coping behavior items, and seven stressor items were assumed to be associated with mild lockdown (Table 1). We asked participants to rate the frequency of implementation and experience of these items from the start of the mild lockdown to the time of the survey on a scale of 1 (not at all) to 7 (extremely). Since these items have not been validated for total score, each item was scored separately.

### 2.3. Statistical Analysis

The *χ*^2^ test was applied to compare sociodemographic data between the three AUDIT groups. The *t*-test was used to compare psychological indexes between the sexes. We used multivariate analysis of variance (MANOVA) and one-way analysis of variance (ANOVA) to compare each variable between the three AUDIT groups, and a post hoc test with Scheffe’s method was employed to test the differences between groups. For all tests, significance was set at *α* = 0.05, two-tailed. Cramer’s *V* (0.100 small; 0.300 medium; 0.600 large) and *η*^2^ (0.010 small; 0.060 medium; 0.140~ large) were calculated as the effect sizes for *the χ*^2^ test and one-way ANOVA and used to interpret the results because the sample size of this study was large. We used multivariate analysis of covariance (MANCOVA) to compare psychological indexes and items related to lifestyle, coping behavior, and stressors related to mild lockdown conditions between the three AUDIT groups adjusting for sociodemographic characteristics. Multinomial logistic regression analyses using the backward stepwise method were conducted to examine the effects of hazardous alcohol use and potential alcoholism on sociodemographic characteristics, psychological index, and items regarding lifestyle, coping behavior, and stressors related to the mild lockdown. Multicollinearity among the independent variables of the final model was checked to assess potential bias in the results due to collinearity. Statistical analyses were performed using SPSS software (version 25.0; IBM Corp, New York, NY, USA).

## 3. Results

### 3.1. Descriptive Results

A total of 11453 individuals participated in the study. Excluding participants under the age of 20, who were prohibited from drinking by Japanese law, 11,427 participants were included in the analysis (48.5% female, mean age = 48.82 ± 13.30 years, range = 20–90 years). In our dataset, although 1152 participants (10.1%) did not provide any data regarding annual household income, there were no missing data for the other variables. The item “Unknown” in annual household income includes the missing values (*N* = 1254).

The average AUDIT scores in males and females were 5.42 ± 6.51 and 2.82 ± 4.80. Based on the classification according to AUDIT scores, the No Problem group (≤ 7) had 9379 (82.1%), the hazardous user group (8–14) had 1221 (10.7%), and the potential alcoholism group (≥15) had 827 (7.2%) (Table 2) people each. There was a significant difference in age between the three groups (*F* [2, 11424] = 21.31, *p* < 0.001, *η*^2^ = 0.004), but the difference did not exceed the lower limit of “small effect size” (i.e., *η*^2^ > 0.010). The age of the hazardous user group (51.05 ± 12.61 years) was significantly higher than those of the No Problem group (48.47 ± 13.52 years, *p* < 0.001) and the Potential Alcoholism group (49.46 ± 11.37 years, *p* = 0.024). Although there were significant differences between the groups in all psychological indexes, only GAD-7 and LSNS-6 exceeded the lower limit of the “small effect size” (Cohen’s *d* > 0.200, Table 3).

### 3.2. Differences in Sociodemographic Characteristics between the Three AUDIT Groups

Table 2 shows the differences in sociodemographic data between the three AUDIT groups. There were significant differences between the three AUDIT groups in all sociodemographic characteristics. The results of the residual analysis that exceeded the lower limit of “small effect size” (i.e., Cramer’s *V* > 0.100) showed that the prevalence of males in the hazardous user and potential alcoholism groups was higher than in the no problem group.

### 3.3. Differences in Psychological Index Scores and COVID-19 Related Variables between the Three AUDIT Groups

In the result of MANOVA for psychological index scores and COVID-19 related variables in the three AUDIT groups, the Box’s M value of 1493.69 was associated with a *p*-value of <0.001, which indicated that the covariance matrices between the groups were not assumed to be equal for the MANOVA. Thus, the one-way ANOVA was conducted to compare the three AUDIT groups.

Table 4 indicates the differences in psychological index scores between the three AUDIT groups. There were significant differences between the three AUDIT groups in all psychological index scores. Regarding the K6, PHQ-9, and GAD-7, which exceeded the lower limit of “small effect size” (i.e., *η*^2^ > 0.010), multiple comparison analysis by Scheffe’s method showed that the scores in the potential alcoholism group were significantly higher than those in the hazardous user and no problem groups.

Table 5 shows the differences in COVID-19 related variables between the three AUDIT groups. There were significant group differences in all items except “Offline interaction with familiar people”. In the items “Altruistic preventive behaviors of COVID-19”, “Deterioration of relationship with familiar people”, “COVID-19-related sleeplessness”, “Difficulties owing to the lack of daily necessities”, and “Difficulties in work or schoolwork”, when the item exceeded the lower limit of “small effect size” (*η*^2^ > 0.010), multiple comparisons using Scheffe’s method were conducted. These indicated that there were significant differences in all combinations between the three AUDIT groups in “Altruistic preventive behaviors of COVID-19”, “Deterioration of relationship with familiar people” and “Difficulties in work or schoolwork”. In other variables that exceeded the lower limit of “small effect size”, the potential alcoholism group showed a significantly higher score than the no problem and hazardous user group.

### 3.4. Psychosocial Factors Relating to Hazardous Alcohol Use and Potential Alcoholism

In results of the MANCOVA for psychological index scores and COVID-19 related variables across the three AUDIT groups adjusting for sex, age group, job classification, marital status, presence of a child, and annual household income, the Box’s M value of 1475.14 was associated with a *p*-value of <0.001, indicating that the covariance matrices between the groups were not assumed to be equal.

Table 6 shows the results of the final multinomial logistic regression analysis examining psychosocial factors relating to hazardous alcohol use and potential alcoholism. No multicollinearity problems were found among the independent variables (all variance inflation factors <4.60). The variables that significantly related to the presence of hazardous alcohol use or the presence of potential alcoholism included male sex, high annual household income, higher age, higher LSNS-6 score, exercise, healthy eating habits, altruistic preventive behaviors of COVID-19, and difficulties in work or schoolwork. Additionally, the variables that significantly related to the presence of potential alcoholism also included higher GAD-7 and PHQ-9 scores, healthy sleeping habits, online interaction with familiar people, deterioration of relationships with familiar people, lower COVID-19-related anxiety, and COVID-19-related sleeplessness.

## 4. Discussion

Our study indicated the prevalence of potential alcoholism (AUDIT score ≥15) and hazardous alcohol use (AUDIT score = 8–14) in the Japanese population during the COVID-19 pandemic. The prevalence of potential alcoholism (men: 10.3%; women: 4.0%) and hazardous alcohol use (men: 14.7%; women: 6.4%) were higher than those found previously in Japan before the pandemic in 2018 (potential alcoholism: 5.2% for men and 0.7% for women; hazardous alcohol use: 16.2% for men and 3.8% for women) [25]. Even though the number of men who engage in problematic drinking has not increased, the number of cases that develop into addiction may have increased. Moreover, depression and anxiety in women participants in this study were relatively higher than those in men, as well as when compared to the results found in many previous studies. This suggests the significance of examining whether women could be more mentally vulnerable to developing hazardous alcohol use or alcoholism in response to the prolonged stressful situation of the COVID-19 pandemic than men. Additionally, differences in the liver and muscle size and gastric emptying rate between men and women make women more susceptible to higher blood alcohol levels [49], and they may be more biologically vulnerable to alcohol-induced brain damage [50,51] and cirrhosis [52]. Although the prevalence of alcohol-related problems among men is a major problem during the COVID-19 pandemic, this study shows that it may be an important issue for women as well. Moreover, a previous study [53] suggested that the cut-off points of the AUDIT for effective detection of hazardous drinking in women need to be lowered from the originally recommended value of 8, and thus, there could be more women with hazardous alcohol use during the pandemic.

The potential alcoholism group scored significantly higher than the other groups on several psychological measures, including psychological distress, depression, and anxiety, while the hazardous user group scored the lowest. The results of multinomial regression analysis showed that higher anxiety and depression related to the presence of potential alcoholism, but not of hazardous alcohol use. These results suggest that it is not necessarily true that the seriousness of the drinking problem does not directly correlate to the seriousness of the psychological problem. Individuals with hazardous alcohol use should receive special notice as their mental problems may be less pronounced; therefore, they may not seek medical attention. In addition, the COVID-19 pandemic is stressful for almost all people, as psychological distress and depression scores in the participants outside of the potential alcoholism group were also higher than the situation before the pandemic [54,55]. Regarding loneliness and social network, while the differences between groups were not prominent considering the ANOVA results, the multinomial regression analysis showed that increased social network size related to the presence of hazardous alcohol use and potential alcoholism. The LSNS-6 does not provide information about the quality of relationships, and thus, if they have poor interpersonal relationships with many people in their social networks, the resulting deterioration in their mental health may affect alcohol use. This speculation needs to be tested in future studies. However, loneliness and social network in our participants were also generally worse than in previous studies before the pandemic [39,40]. This suggests that those with addictive drinking habits may be at an even greater risk of worsening symptoms of alcoholism during the pandemic and that there might be an increased risk of developing alcoholism in non-addictive alcohol users with psychological vulnerability.

Regarding the results of the ANOVA for COVID-19-related variables, decreased altruistic preventive behaviors of COVID-19, deterioration of relationships with familiar people, COVID-19-related sleeplessness, difficulties due to the lack of daily necessities, and difficulties in work or schoolwork were prominent in the potential alcoholism group. Multinomial logistic regression analysis showed that these variables, except difficulties due to the lack of daily necessities, exercise, decreased healthy eating habits, decreased healthy sleeping habits, online interaction with familiar people, and decreased COVID-19-related anxiety related to the presence of potential alcoholism. Various difficulties in daily life due to the COVID-19 pandemic associated with the presence of potential alcoholism and individuals with it may require support in these issues. It was suggested that even if online interactions with familiar people increase, poor relationships with them may lead to worsening drinking behavior. The interpersonal relationship was previously reported to be a serious problem during the COVID-19 pandemic [56]. While drinking behavior is promoted as a way to cope with such interpersonal stress, it has also been reported that drinking as a way to cope with stress can increase interpersonal stress [57]. It was also suggested that not only relationship problems, but also COVID-19-related sleep problems could be targets for interventions for potential alcoholism. Although poor eating and sleep habits were also risk factors for potential alcoholism, increased exercise related to its presence. It is not difficult to imagine that a poor lifestyle may lead to potential alcoholism, but the background of the results regarding exercise is difficult to interpret and requires further examination. While difficulties due to the lack of daily necessities and difficulties in work in people with potential alcoholism were prominent, and the latter related to the presence of potential alcoholism, they did not have a higher prevalence in the low-income group than in other groups, suggesting that these problems were not due to economic reasons, and a detailed analysis of the causes of these problems is needed in the future. The decrease in altruistic preventive behaviors and the disturbance of sleep due to concern about COVID-19 infection in people with potential alcoholism appear to be contradictory, but the latter might be caused by a decrease in offline interaction with others due to deteriorating relationships, and they may have fewer opportunities for altruistic preventive behaviors. The result that decreased COVID-19-related anxiety is associated with the presence of potential alcoholism is difficult to interpret. The item “I felt nervous or anxious when I watched the news about coronavirus disease 2019” is limited to situations where people are watching the news about COVID-19, and thus, people with potential alcoholism may not have access to (or avoid) such news. Regarding hazardous alcohol use, exercise, decreased healthy eating habits, altruistic preventive behaviors of COVID-19, and difficulties in work or schoolwork were risk factors. The addition of these factors to various psychosocial factors found only in potential alcoholism could exacerbate the problem of drinking. The same is true for psychological variables, but effect size is generally lower in the comparison between the three AUDIT groups for the COVID-19-related variables (see Table 4 and Table 5). This may be due in part to the strong influence of demographic characteristics on the dependent variable, which increased the variance of values in each group. Analysis adjusting demographic characteristics may be essential to explore the relationship between alcohol use and psychological indexes or items regarding lifestyle, coping behavior, and stressors related to mild lockdown conditions (i.e., multinomial regression analysis or MANCOVA, which were conducted in this study).

This study had several limitations. First, we did not collect data before the COVID-19 pandemic, and thus, we could not assess changes in alcohol use during the COVID-19 pandemic. Second, the data for this study were collected through an online survey and random sampling could not be conducted, so the representativeness of the sample could not be guaranteed; the sample we collected could not be matched to the proportions of each age group and gender group in each region. Third, because the study participants were registered with a survey company, they were likely to be more accustomed to completing surveys than individuals who were not registered, and the motivation to cooperate with the survey may differ between our participants and the population not registered with the survey company. Fourth, the items to investigate lifestyle, coping behavior, and stressors related to mild lockdown were developed in this study based on previous research. Therefore, the reliability and validity of these items should be examined in future studies.

## 5. Conclusions

This study investigated alcohol use during the prolonged COVID-19 pandemic and its related psychosocial and demographic characteristics in the Japanese population. We found that the prevalence of potential alcoholism during the COVID-19 pandemic in our study was higher than that reported in previous studies, and this trend was particularly pronounced among women. The presence of potential alcoholism was related with by a deteriorated psychological status, particularly depression and anxiety. Psychological variables in our participants deteriorated in comparison with previous results before the pandemic, and thus, there could be an increased risk of worsening or developing alcoholism during the pandemic. Various difficulties in daily life due to the pandemic also related to the presence of potential alcoholism, and individuals with it may require support in these issues. In the future, intervention methods and systems should be developed to provide optimal assistance to people with psychological problems who are vulnerable to alcohol-related problems during the COVID-19 pandemic while conducting further long-term follow-up studies.

## Figures and Tables

**Table 1 ijerph-18-13318-t001:** Items about lifestyle, coping behavior, and stressors related to mild lockdown.

1.	I exercised for my health (whether indoors or outdoors).
2.	I took meals considering the nutrition balance.
3.	I kept regular awakening time and bedtime approximately.
4.	I engaged in activities such as hobbies with absorbing interest.
5.	I interacted with my family or friends on a face-to-face basis (outside of work or class).
6.	I interacted with my family or friends online using chat or video calling (except work or class).
7.	I spontaneously refrained from going out or took preventive behaviors (e.g., wearing a mask) to prevent coronavirus disease 2019 infection to my family or other people.
8.	I thought about the future positively.
9.	The family budget has tightened.
10.	A personal relationship with a close person such as family or friends got worse.
11.	I have become easily annoyed or irate due to life changes.
12.	I felt nervous or anxious when I watched the news about coronavirus disease 2019.
13.	Because I kept thinking about coronavirus disease 2019 infection, I could not sleep.
14.	My daily life was disrupted due to the shortage of materials relating to prevention for coronavirus disease 2019 infection (e.g., mask or thermometer) or other daily supplies.
15.	My work or schoolwork was disrupted due to a life change.

**Table 2 ijerph-18-13318-t002:** Comparisons of sociodemographic data between three AUDIT group.

		*N* (%) in Each AUDIT Group						Group Difference		
	Total	No Problem		Hazardous User		Potential Alcoholism		*χ* ^2^	*p*	Cramer’s *V*
Overall	11,427	9379	(82.1)	1221	(10.7)	827	(7.2)			
Sex								409.01	<0.001	0.189
*Male*	5881	4413	(75.0) [−]	865	(14.7) [+]	603	(10.3) [+]			
*Female*	5546	4966	(89.5) [+]	356	(6.4) [−]	224	(4.0) [−]			
Age								108.40	<0.001	0.069
*20–39*	3018	2634	(87.3) [+]	229	(7.6) [−]	155	(5.1) [−]			
*40–64*	6922	5500	(79.5) [−]	817	(11.8) [+]	605	(8.7) [+]			
*≥65*	1487	1245	(83.7)	175	(11.8)	67	(4.5) [−]			
Occupation								156.61	<0.001	0.083
*Employed*	7994	6360	(79.6) [−]	949	(11.9) [+]	685	(8.6) [+]			
*Homemaker*	1754	1609	(91.7) [+]	98	(5.6) [−]	47	(2.7) [−]			
*Student*	124	102	(82.3)	13	(8.8)	9	(7.3)			
*Unemployed*	1213	1023	(84.3) [+]	124	(10.2)	66	(5.4) [−]			
*Other*	342	285	(83.3)	37	(10.8)	20	(5.8)			
Marital status								14.67	0.001	0.036
*Married*	7217	5864	(81.3) [−]	832	(11.5) [+]	521	(7.2)			
*Unmarried*	4210	3515	(83.5) [+]	389	(9.2) [−]	306	(7.3)			
The presence of child								12.42	0.002	0.033
*Yes*	6388	5185	(81.2) [−]	740	(11.6) [+]	463	(7.2)			
*No*	5039	4194	(83.2) [+]	481	(9.5) [−]	364	(7.2)			
Annual household income								67.22	<0.001	0.061
*<2.0 million JPY*	715	601	(84.1) [+]	57	(8.0) [−]	57	(8.0)			
*2.0–3.9 million JPY*	2065	1740	(84.3) [+]	207	(10.0) [−]	118	(5.7) [−]			
*4.0–5.9 million JPY*	2246	1819	(81.0)	271	(12.1)	156	(6.9)			
*6.0–7.9 million JPY*	1670	1343	(80.4)	178	(10.7)	149	(8.9) [+]			
*≥8.0 million JPY*	2313	1757	(76.0) [−]	339	(14.7) [+]	217	(9.4) [+]			

Cramer’s *V*: 0.100~small; 0.300~medium; 0.600~large. [+]: adjusted residuals ≥ 1.96 [−]: adjusted residuals ≤ −1.96. AUDIT: Alcohol Use Disorders Identification Test. No Problem: AUDIT score < 8. Hazardous User: AUDIT score = 8~14. Potential Alcoholism: AUDIT score ≥ 15.

**Table 3 ijerph-18-13318-t003:** Comparisons of psychological index scores between men and women.

	Mean (SD)				Group Difference			
	Male		Female		Difference (95%CI)		*p*	Cohen’s *d*
K6	3.77	(5.26)	4.78	(5.50)	−1.00	(−1.20, −0.80)	<0.001	0.186
PHQ-9	3.76	(5.62)	4.83	(5.79)	−1.06	(−1.27, −0.85)	<0.001	0.187
GAD-7	2.71	(4.40)	3.69	(4.79)	−0.98	(−1.15, −0.81)	<0.001	0.213
UCLA-LS3	24.37	(5.60)	23.71	(6.08)	0.66	(0.45, 0.87)	<0.001	0.113
LSNS-6	8.92	(6.30)	10.25	(5.81)	−1.33	(−1.55, −1.11)	<0.001	0.219

Cohen’s *d*: 0.200~small; 0.500~medium; 0.800~large.

**Table 4 ijerph-18-13318-t004:** Comparisons of psychological index scores between three AUDIT groups.

	Mean (SD)								Group Difference		
	Total		No Problem		Hazardous User		Potential Alcoholism		*F*	*p*	*η* ^2^
K6	4.26	(5.40)	4.08	(5.29)	3.90	(4.95)	6.83	(6.49)	103.36	<0.001 *1	0.018
PHQ-9	4.28	(5.72)	4.05	(5.53)	3.93	(5.31)	7.36	(7.39)	132.54	<0.001 *1	0.023
GAD-7	3.19	(4.62)	3.01	(4.48)	2.89	(4.16)	5.64	(5.97)	128.23	<0.001 *1	0.022
UCLA-LS3	24.05	(5.85)	24.02	(5.92)	23.33	(5.57)	25.40	(5.11)	31.70	<0.001 *2	0.006
LSNS-6	9.57	(6.11)	9.43	(6.03)	10.51	(6.32)	9.66	(6.50)	16.82	<0.001 *3	0.003

*η*^2^: 0.010~ small; 0.060 medium; 0.140~ large. SD: standard deviation. No Problem: AUDIT score < 8. Hazardous User: AUDIT score = 8~14. Potential Alcoholism: AUDIT score ≥ 15. AUDIT: Alcohol Use Disorders Identification Test. K6: Kessler Psychological Distress Scale-6. PHQ-9: Patient Health Questionnaire-9. GAD-7: Generalized Anxiety Disorder-7. UCLA-LS3: UCLA Loneliness Scale Version 3. LSNS-6: the abbreviated Lubben Social Network Scale. *1 Significant difference between the no problem group and the potential alcoholism group and between the hazardous user group and the potential alcoholism group in multiple comparison analysis. *2 Significant differences between all groups in multiple comparison analyses. *3 Significant difference between the no problem group and the hazardous user group and between the hazardous user group and the potential alcoholism group in multiple comparison analysis.

**Table 5 ijerph-18-13318-t005:** Comparisons of COVID-19 related variables between three AUDIT groups.

	Mean (SD)								Group Difference		
	Total		No Problem		Hazardous User		Potential Alcoholism		*F*	*p*	*η* ^2^
Exercise	3.67	(1.89)	3.62	(1.90)	3.93	(1.88)	3.85	(1.84)	18.64	<0.001 *1	0.003
Healthy eating habits	4.13	(1.62)	4.15	(1.62)	4.15	(1.60)	3.93	(1.59)	6.86	0.001 *2	0.001
Healthy sleep habits	4.64	(1.74)	4.67	(1.74)	4.69	(1.69)	4.18	(1.68)	31.76	<0.001 *2	0.006
Favorite activity	3.65	(1.69)	3.63	(1.70)	3.81	(1.69)	3.67	(1.61)	6.80	0.001 *3	0.001
Offline interaction with familiar people	3.36	(1.79)	3.34	(1.80)	3.43	(1.78)	3.50	(1.67)	2.87	0.057	0.001
Online interaction with familiar people	2.66	(1.76)	2.61	(1.75)	2.73	(1.74)	3.08	(1.76)	27.78	<0.001 *2	0.005
Altruistic preventive behaviors of COVID-19	5.41	(1.70)	5.47	(1.69)	5.29	(1.72)	4.82	(1.77)	59.05	<0.001 *4	0.010
Optimism	4.15	(1.56)	4.16	(1.57)	4.28	(1.47)	3.88	(1.58)	16.65	<0.001 *4	0.003
Deterioration of household economy	3.49	(1.73)	3.45	(1.73)	3.49	(1.71)	3.90	(1.75)	25.74	<0.001 *2	0.004
Deterioration of relationship with familiar people	2.64	(1.58)	2.58	(1.56)	2.72	(1.60)	3.26	(1.66)	72.68	<0.001 *4	0.013
Frustration	3.23	(1.73)	3.19	(1.73)	3.21	(1.67)	3.71	(1.72)	35.56	<0.001 *2	0.006
COVID-19-related anxiety	3.47	(1.68)	3.45	(1.68)	3.41	(1.63)	3.72	(1.70)	10.26	<0.001 *2	0.002
COVID-19-related sleeplessness	2.47	(1.52)	2.41	(1.50)	2.46	(1.49)	3.11	(1.69)	79.99	<0.001 *2	0.014
Difficulties owing to the lack of daily necessities	2.57	(1.57)	2.50	(1.55)	2.60	(1.54)	3.23	(1.71)	82.58	<0.001 *2	0.014
Difficulties in work or schoolwork	2.86	(1.73)	2.78	(1.71)	2.96	(1.75)	3.50	(1.78)	67.71	<0.001 *4	0.012

*η*^2^: 0.010~ small; 0.060 medium; 0.140~ large. SD: standard deviation. No Problem: AUDIT score < 8. Hazardous User: AUDIT score = 8~14. Potential Alcoholism: AUDIT score ≥15. AUDIT: Alcohol Use Disorders Identification Test. COVID-19: Coronavirus disease 2019. *1 Significant difference between the no problem and hazardous user groups and between the no problem and potential alcoholism groups in the multiple comparison group. *2 Significant difference between the no problem and the potential alcoholism group and between the hazardous user and potential alcoholism groups in multiple comparison analysis. *3 Significant difference between the no problem and hazardous user groups in the multiple comparison analysis. *4 Significant differences between all groups in multiple comparison analyses.

**Table 6 ijerph-18-13318-t006:** Results of multinomial logistic regression analysis.

	Hazardous Use			Potential Alcoholism		
Independent Variables	OR (95% CI)		*p*	OR (95% CI)		*p*
Sex (ref: female)						
*Male*	2.38	(2.00–2.84)	<0.001	2.29	(1.85–2.84)	<0.001
Occupation (ref: other)						
*Employed*	1.14	(0.75–1.73)	0.545	1.39	(0.81–2.39)	0.231
*Homemaker*	0.89	(0.55–1.45)	0.642	0.80	(0.42–1.53)	0.500
*Student*	1.52	(0.57–4.06)	0.402	1.52	(0.45–5.09)	0.498
*Unemployed*	0.91	(0.58–1.45)	0.699	0.82	(0.45–1.52)	0.534
Annual household income (ref: ≥ 8.0 million)						
*6.0–7.9 million*	0.74	(0.61–0.90)	0.003	0.95	(0.75–1.20)	0.653
*4.0–5.9 million*	0.88	(0.73–1.05)	0.154	0.78	(0.62–0.98)	0.030
*2.0–3.9 million*	0.74	(0.61–0.91)	0.003	0.61	(0.47–0.79)	<0.001
*<2.0 million*	0.61	(0.44–0.84)	0.003	0.71	(0.50–1.01)	0.055
Age	1.01	(1.00–1.02)	0.004	1.02	(1.01–1.03)	<0.001
PHQ-9	1.02	(1.00–1.04)	0.101	1.05	(1.02–1.07)	<0.001
GAD-7	1.00	(0.97–1.03)	0.866	1.05	(1.02–1.09)	<0.001
UCLA	0.99	(0.97–1.00)	0.089	1.01	(0.99–1.03)	0.200
LSNS-6	1.03	(1.02–1.05)	<0.001	1.04	(1.02–1.06)	<0.001
Exercise	1.08	(1.03–1.13)	0.001	1.13	(1.06–1.20)	<0.001
Healthy eating habits	0.94	(0.89–1.00)	0.038	0.88	(0.82–0.95)	<0.001
Healthy sleep habits	1.00	(0.95–1.05)	0.946	0.93	(0.88–0.99)	0.018
Online interaction with familiar people	1.00	(0.95–1.04)	0.879	1.11	(1.05–1.18)	<0.001
Altruistic preventive behaviors of COVID-19	0.94	(0.90–0.99)	0.010	0.86	(0.81–0.91)	<0.001
Deterioration of relationship with familiar people	1.06	(1.00–1.12)	0.055	1.09	(1.01–1.16)	0.018
COVID-19-related anxiety	1.02	(0.96–1.07)	0.597	0.91	(0.84–0.98)	0.012
COVID-19-related sleeplessness	0.95	(0.89–1.01)	0.090	1.10	(1.02–1.19)	0.015
Difficulties in work or schoolwork	1.05	(1.00–1.11)	0.040	1.09	(1.02–1.16)	0.010

Note. *R*^2^ = 0.095 (Cox–Snell), and 0.133 (Nagelkerke). Model *χ*^2^(46) = 898.94, *p* < 0.001. No Problem: AUDIT score < 8. Hazardous Use: AUDIT score = 8 − 14. Potential Alcoholism: AUDIT score 15. COVID-19: Coronavirus disease 2019. PHQ-9: Patient Health Questionnaire-9. GAD-7: Generalized Anxiety Disorder-7. UCLA-LS3: UCLA Loneliness Scale Version 3. LSNS-6: the abbreviated Lubben Social Network Scale.

## Data Availability

The data presented in this study are available on request from the corresponding author.

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
