# Peer review of "Alcohol Use and Its Related Psychosocial Effects during the Prolonged COVID-19 Pandemic in Japan: A Cross-Sectional Survey"

_ijerph, 2021, doi:10.3390/ijerph182413318_

Round 1

Reviewer 1 Report

This is a study of interest, although after reading it I have raised different questions and questions that I ask the authors to answer.

INTRODUCTION

In relation to changes in alcohol consumption during COVID-19, it is suggested to review different studies published in this regard that provide more information on the influence of variables such as sex, age and other sociodemographic variables that are considered in the manuscript. They can enrich both the Introduction and the Discussion (i.e. in the special issue on Alcohol of this same journal there are some studies).

A literature review is needed to justify the inclusion of psychosocial variables considered in the Method section. They are hardly addressed, rather than superficial, and not all. This aspect needs to be more comprehensively completed.

METHOD

In the Method section, when describing the AUDIT, it should be noted whether there are differences according to sex, as considered in other AUDIT validations. This aspect is key to interpreting and discussing findings based on sex.

The questionnaire "Lifestyle, Coping Behavior, and Stressors Related to the Mild Lockdown" should be noted that it is not validated and that, consequently, it may present some limitations in the interpretation of the findings that refer to this.

DISSCUSION

"The average AUDIT scores in males and females were 5.42±6.51 and 2.82±4.80". These results may be influenced by the failure to take into account differences in the interpretation according to sex of the scores obtained in the AUDIT. He could be underestimated in the case of women. Additionally, for some of the analyses that have been developed in relation to sociodemographic characteristics, it should be taken into account to perform them by controlling the sex variable, or by performing an Analysis of Variance (ANOVA) to study the effect of interaction.

"Our study indicated that the prevalence of potential alcoholism (AUDIT score ≥ 15) 252 and hazardous alcohol use (AUDIT score = 8–14) in the Japanese population increased 253 during the COVID-19 pandemic". This statement cannot be made, as they do not make a comparative study between the consumption immediately prior to, and during, the pandemic. In fact, the authors themselves refer to it in Limitations of their study. The prevalence data presented to establish this claim correspond to a study carried out 7 years earlier. As a result, the entire structure of the Discussion, compared to this 2013 study, is unsuitable for establishing statements about changes in alcohol consumption during the pandemic. The observed increase may have occurred during those 7 years. Authors are suggested to use epidemiological sources from 2019 or the first 2 months of 2020. All statements made regarding the increase in the number of cases during the pandemic, etc., should be reviewed. 

Another option is to make a comparison with the study cited by the authors in the Introduction: Sato, K.; Kobayashi, S.; Yamaguchi, M.; Sakata, R.; Sasaki, Y.; Murayama, C.; Kondo, N. Working from Home and Dietary 362 Changes during the COVID-19 Pandemic: A Longitudinal Study of Health App (Calo Mama) Users. Appetite 2021, 165, 105323. 363 https://doi.org/10.1016/j.appet.2021.105323.

Alternatively, don't make such a comparison, and focus on presenting the prevalence of each level of alcohol consumption in the evaluation period, without indicating whether or not there have been increases or decreases in consumption compared to before the pandemic. In this sense, focus the Discussion on the findings that relate alcohol consumption with the other variables evaluated.

This is also seen with other study variables, which are also said to have decreased or increased during the pandemic. The Discussion should be reviewed and limited to the relationship between alcohol consumption and alcohol consumption from a transversal perspective, specified in the period evaluated.

"This shows that women could be more mentally vulnerable to developing hazardous 266 alcohol use or alcoholism in response to the prolonged stressful situation of the COVID-267 19 pandemic than men". This statement cannot be made as it is a cross-sectional study. One can only point out the relationships between alcohol consumption and anxiety, but not speak of vulnerability before situations of prolonged stress by the COVID.

CONCLUSIONS

In view of the above, the Conclusions should be revised.

Author Response

Dear Reviewer 1,

We appreciate the constructive comments provided. We have revised the manuscript considering your suggestions, and the revised text is indicated in red.

Comment 1:

(INTRODUCTION) In relation to changes in alcohol consumption during COVID-19, it is suggested to review different studies published in this regard that provide more information on the influence of variables such as sex, age and other sociodemographic variables that are considered in the manuscript. They can enrich both the Introduction and the Discussion (i.e. in the special issue on Alcohol of this same journal there are some studies). A literature review is needed to justify the inclusion of psychosocial variables considered in the Method section. They are hardly addressed, rather than superficial, and not all. This aspect needs to be more comprehensively completed.

Response:

As you pointed out, the review of previous studies in the Introduction section was insufficient. Thus, we added previous studies and described the demographic characteristics and psychological factors associated with alcohol use during the coronavirus (COVID-19) pandemic in more detail. (Lines 45–61)

Comment 2:

(METHOD) In the Method section, when describing the AUDIT, it should be noted whether there are differences according to sex, as considered in other AUDIT validations. This aspect is key to interpreting and discussing findings based on sex.

Response:

As advised, we added the explanation that previous research conducted in the Japanese population before the COVID-19 pandemic reported that men showed higher AUDIT scores than women (Osaki et al., 2016). (Lines 131–133)

Furthermore, a previous study (Reinert et al., 2007) suggested that the cut-off points for effective detection of hazardous drinking in women need to be lowered from the originally recommended value of 8, and for this reason, there could be more women with hazardous alcohol use during the pandemic. We also added an explanation regarding this possibility to the Discussion section. (Lines 298–301)

Comment 3:

(METHOD) The questionnaire "Lifestyle, Coping Behavior, and Stressors Related to the Mild Lockdown" should be noted that it is not validated and that, consequently, it may present some limitations in the interpretation of the findings that refer to this.

Response:

As you pointed out, the items to investigate lifestyle, coping behavior, and stressors related to mild lockdown were developed in this study based on previous research. Therefore, the reliability and validity of these items should be examined in future studies. We added this problem as the fourth limitation to this study. (Lines 372–375)

Comment 4:

(DISSCUSION) "The average AUDIT scores in males and females were 5.42±6.51 and 2.82±4.80". These results may be influenced by the failure to take into account differences in the interpretation according to sex of the scores obtained in the AUDIT. He could be underestimated in the case of women. Additionally, for some of the analyses that have been developed in relation to sociodemographic characteristics, it should be taken into account to perform them by controlling the sex variable, or by performing an Analysis of Variance (ANOVA) to study the effect of interaction.

Response:

Other reviewers made similar comments about controlling for demographic data such as sex; thus we conducted a multinomial logistic regression analysis based on these comments. (Lines 264–275, Table 6)

Comment 5:

(DISSCUSION) "Our study indicated that the prevalence of potential alcoholism (AUDIT score ≥ 15) and hazardous alcohol use (AUDIT score = 8–14) in the Japanese population increased during the COVID-19 pandemic". This statement cannot be made, as they do not make a comparative study between the consumption immediately prior to, and during, the pandemic. In fact, the authors themselves refer to it in Limitations of their study. The prevalence data presented to establish this claim correspond to a study carried out 7 years earlier. As a result, the entire structure of the Discussion, compared to this 2013 study, is unsuitable for establishing statements about changes in alcohol consumption during the pandemic. The observed increase may have occurred during those 7 years. Authors are suggested to use epidemiological sources from 2019 or the first 2 months of 2020. All statements made regarding the increase in the number of cases during the pandemic, etc., should be reviewed.

Another option is to make a comparison with the study cited by the authors in the Introduction: Sato, K.; Kobayashi, S.; Yamaguchi, M.; Sakata, R.; Sasaki, Y.; Murayama, C.; Kondo, N. Working from Home and Dietary Changes during the COVID-19 Pandemic: A Longitudinal Study of Health App (Calo Mama) Users. Appetite 2021, 165, 105323. https://doi.org/10.1016/j.appet.2021.105323.

Alternatively, don't make such a comparison, and focus on presenting the prevalence of each level of alcohol consumption in the evaluation period, without indicating whether or not there have been increases or decreases in consumption compared to before the pandemic. In this sense, focus the Discussion on the findings that relate alcohol consumption with the other variables evaluated.

This is also seen with other study variables, which are also said to have decreased or increased during the pandemic. The Discussion should be reviewed and limited to the relationship between alcohol consumption and alcohol consumption from a transversal perspective, specified in the period evaluated.

Response:

Thank you for your detailed suggestions. As you pointed out, we cannot rule out the possibility that the comparison with the 2013 survey (Osaki et al., 2016) has changed over the long term, so we removed the statement about the comparison with this previous study. However, another study (Kinjo et al., 2019) was conducted in 2018, and we considered it was suitable for comparison. Therefore, we added more details about this previous study and clearly stated that the survey was conducted in 2018 (Lines 282–285). In addition, we refrained from using categorical statements when describing this comparison (Line 289–290). The first sentence in the first paragraph of the discussion has been changed to state that the results of this study show the status of alcohol use problems, not an increase in alcohol use during the pandemic (Line 280–282).

Comment 6:

(DISSCUSION) "This shows that women could be more mentally vulnerable to developing hazardous alcohol use or alcoholism in response to the prolonged stressful situation of the COVID-19 pandemic than men". This statement cannot be made as it is a cross-sectional study. One can only point out the relationships between alcohol consumption and anxiety, but not speak of vulnerability before situations of prolonged stress by the COVID.

Response:

As you pointed out, vulnerability should not be mentioned categorically. We rephrased this segment as follows:

This suggests the significance of examining whether women could be more… (Line 289–290).

Comment 7:

(CONCLUSIONS) In view of the above, the Conclusions should be revised.

Response:

Based on the revised results, we have rewritten the conclusion section. (Lines 381–386)

Reviewer 2 Report

This study evaluates the association between alcohol use and related psychosocial effects during the COVID 19 pandemic in Japan. Overall, a well-executed study with large sample sizes. Having said that, I hope the authors will consider my suggestion below on the statistical methodology, before they draw a final conclusion.

  1. Line 52-54: can you cite this ref/study that could potentially be helpful: https://sciforum.net/manuscripts/9018/manuscript.pdf
  2. Line 187: What you had in table 2 was great; but could you please consider running a logistic regression of alcohol use (no problem, hazardous user and potential alcoholism) on all the socio demographics correlates – all in a single model? Reason being is because the current analysis have not adjusted for potential confound and your discussion was around gender. I think these observed results in table 2 could be inflated.
  3. Ditto for table 3, you should first check the correlation between each of these psychological index scores. I think k6 will be highly correlated with PHQ 9 and GAD7. Then repeat the logistic regression model above (in point #2) and the psychological index scores. If some of the index scores are highly correlated, then exclude them in the regression model.
  4. Line 237: table 4; Again you have the option of repeating the methods I suggested in (2) & (3) or you can run a factor analysis to find some latent variables and then run a logistic regression.
  5. In those logistic regression above, if you are comparing non-user vs hazardous, non-user vs potential alcoholism and hazardous versus potential alcoholism, then you will need to lower the significance level to adjust for multiple comparison.

Author Response

Dear Reviewer 2

We appreciate the constructive comments provided. We have revised the manuscript accordingly. The revised text is indicated in red.

Comment 1:

Line 52-54: can you cite this ref/study that could potentially be helpful: https://sciforum.net/manuscripts/9018/manuscript.pdf

Response:

    Thank you for your suggestion. The paper you recommended is very useful in supporting my description. Thus, we cited this paper. (Line 68)

Comment 2:

(Point #2) Line 187: What you had in table 2 was great; but could you please consider running a logistic regression of alcohol use (no problem, hazardous user and potential alcoholism) on all the socio demographics correlates – all in a single model? Reason being is because the current analysis have not adjusted for potential confound and your discussion was around gender. I think these observed results in table 2 could be inflated.

(Point #3) Ditto for table 3, you should first check the correlation between each of these psychological index scores. I think k6 will be highly correlated with PHQ 9 and GAD7. Then repeat the logistic regression model above (in point #2) and the psychological index scores. If some of the index scores are highly correlated, then exclude them in the regression model.

(Point #4) Line 237: table 4; Again you have the option of repeating the methods I suggested in (2) & (3) or you can run a factor analysis to find some latent variables and then run a logistic regression.

(Point #5) In those logistic regression above, if you are comparing non-user vs hazardous, non-user vs potential alcoholism and hazardous versus potential alcoholism, then you will need to lower the significance level to adjust for multiple comparison.

Response:

    Thank you very much for your constructive comments. Considering your advice (Points 2–5), we conducted a logistic regression analysis conducted using demographic data, psychological indexes, and items of Lifestyle, Coping Behavior, and Stressors Related to the Mild Lockdown as explanatory variables and AUDIT groups as dependent variables. Since the inter-variable correlations were <0.9, we treated them all as explanatory variables, and also checked for multicollinearity based on the VIF to ensure that there were no problems in using these variables. (Lines 264–275, Table 6) 

Reviewer 3 Report

Dear Authors,

Dear Editor,

The study covers important issues of the lifestyle changes following the pandemic outbreak. However, the literature review does not introduce the multi-level impact of alcohol use and addiction and it is less supportive to the rational for the study and its importance. I strongly recommend to perform statistical analysis for between-groups differences with control for demographic data to allow drawing conclusions on alcohol use impact. The statistical analysis will have substantial impact on the results and discussion, thus these two sections should be re-organized in the next version of the manuscript.

My specific comments are detailed below.

Abstract:

The conclusion on causality between alcohol use and mental health issues (such as psychological distress ad depression), functional issues, etc. was not established in your study since it was a cross-sectional one. For this reason, caution is needed in conclusion on focus for future interventions. It’s possible that the required intervention should address psychological state of the population and functional issues which were caused by the Pandemic, rather than addressing directly alcohol use patterns.  

Introduction:

In general, the introduction should be expanded with materials on general multi-level impact of alcohol use and addiction. Otherwise, the rational for the study and its importance are unclear.

Lines 37-47: Please state clearly that the reported studies covered other than Japan countries.

Lines 58-59: Short report on the findings from early study on alcohol use in Japan during COVID 19 (ref num. 6) is needed to provide background and justification for the current research project.

Materials and Methods

Line 70: The inclusion criteria for the study is not clear. You report that an inclusion criteria was “ caution “. However, in the results section you provide range of ages: age = 48.82±13.30 yrs., range = 20–90 years. Please explain the inclusion criteria.

Line 74: “ Through Macromill.inc. (Tokyo, Japan), a global marketing research company…” are there some specific characteristics for participants of this company in comparison to general population?

Line 97: Please provide reference for cut-of scores for the Alcohol Use Disorders Identification Test (AUDIT) (at the end of the paragraph).

Lines 147-153: You developed a new tool for assessment of Lifestyle, Coping Behavior, and Stressors Related. The description of the development process should be described in more details, including background and initial steps for psychometric properties establishment. At least data on face validity should be presented.

Results

The groups were different as for the number of participants: 82%, 11%, 7% of the total sample size. Please describe statistical procedures that were applied through the analysis to address this difference in the groups’ size.

Lines 175-176: You report on statistical significant differences in age between the groups. However, based on effect size magnitude, this difference has no practical significance. Please elaborate on this issue.  

“Comparisons of psychological index scores between three AUDIT groups” – data which presented in the Table 3 – the statistical analysis is not sufficient to provide answer on the research question. More specifically, there are so many demographic differences between the groups, the analysis on simple difference between the groups does not allow to conclude on the source of the differences. For example, the between group differences in the depression scores may be attributed to the age differences rather than to the alcohol usage patterns. You should control through the analysis at least the main variables of age and gender.

“Comparisons of COVID-19 related variables between three AUDIT groups” – data presented in the table 4 – Again, the same issue of demographic between groups differences affected the results. Moreover, as it was described, the items of COVID-19 related variables were scored on the ordinal scale. Such type of the data does not fit the assumptions of one-way ANOVA analysis, which makes it even more difficult to conclude on something.

Lines 263-265: “Moreover, depression and anxiety in women participants in this study were relatively higher than those in men, as well as when compared to the results found in many previous studies (Appendix A)” – If this information is so important to be discussed first, you should provide the data analysis in the manuscript body instead of appendix.

Lines 274-288: the discussion on the between-groups differences in psychologic indices is less relevant until you re-do analysis with control for the between-groups differences in demographic data.

Lines 289: The differences in the COVID-19 measurements – since (a) the process of development of the tool is not evident through the manuscript and (b) the applied statistical approach seems to be less relevant, the results can be seen as unsupported by the data.

Author Response

Reviewer 3

We appreciate the important comments you provided. We revised the manuscript according to your suggestions. The revised text is indicated in red.

Comment 1:

(Abstract) The conclusion on causality between alcohol use and mental health issues (such as psychological distress ad depression), functional issues, etc. was not established in your study since it was a cross-sectional one. For this reason, caution is needed in conclusion on focus for future interventions. It’s possible that the required intervention should address psychological state of the population and functional issues which were caused by the Pandemic, rather than addressing directly alcohol use patterns. 

Response:

As per your advice, we revised the sentence in the abstract and conclusion section as follows:

In the future, intervention methods and systems should be developed to provide optimal assistance to people with psychological problems who are vulnerable to alcohol-related problems during the COVID-19 pandemic, while conducting further long-term follow-up studies (Lines 28; 388–389)

Comment 2:

(Introduction) In general, the introduction should be expanded with materials on general multi-level impact of alcohol use and addiction. Otherwise, the rational for the study and its importance are unclear.

Response:

As you pointed out, the review of previous studies in the Introduction section was insufficient. Thus, we added previous studies and described the demographic characteristics and psychological factors associated with alcohol use during the pandemic in more detail. (Lines 45–61)

Comment 3:

(Introduction) Lines 37-47: Please state clearly that the reported studies covered other than Japan countries.

Response:

As you suggested, we have specified the countries these previous studies were conducted in. (Lines 39–61)

Comment 4:

(Introduction) Lines 58-59: Short report on the findings from early study on alcohol use in Japan during COVID 19 (ref num. 6) is needed to provide background and justification for the current research project.

Response:

A previous study (Sato et al., 2021) on dietary patterns, including alcohol use in the Japanese population, was conducted during the early stages of the COVID-19 pandemic (January–May 2020), and it reported that their alcohol intake decreased. However, no previous research has investigated problematic alcohol use during the prolonged COVID-19 pandemic and its demographic and psychosocial factors in Japan using large-scale samples, to the best of our knowledge. Moreover, a survey conducted later in the pandemic, such as 1 year after the start of a pandemic, may yield different results from the previous studies described above. Therefore, we have revised this paragraph. (Lines 72–79)

Comment 5:

(Materials and Methods) Line 70: The inclusion criteria for the study is not clear. You report that an inclusion criteria was “ caution “. However, in the results section you provide range of ages: age = 48.82±13.30 yrs., range = 20–90 years. Please explain the inclusion criteria.

Response:

The inclusion criteria were as follows: (a) inhabitants living in the six prefectures (Tokyo, Aichi, Osaka, Kyoto, Hyogo, and Fukuoka) and (b) aged ≥20years. We added this sentence. (Lines 86–88)

Comment 6:

(Materials and Methods) Line 74: “Through Macromill.inc. (Tokyo, Japan), a global marketing research company…” are there some specific characteristics for participants of this company in comparison to general population?

Response:

Since the participants of this study were registered with a survey company, they were likely to be more accustomed to completing surveys than those who were not registered. Additionally, the motivation to cooperate with the survey may differ between our participants and the population not registered with the survey company. Although the effect of this is unknown, we have added this issue as one of the limitations. (Lines 368–372)

Comment 7:

(Materials and Methods) Line 97: Please provide reference for cut-of scores for the Alcohol Use Disorders Identification Test (AUDIT) (at the end of the paragraph).

Response:

We referred to the cut-off criteria of the AUDIT in specific health guidance by the Japanese Ministry of Health, Labour and Welfare and the World Health Organization’s guidelines. We added these references. (Lines 128–130)

Comment 8:

(Materials and Methods) Lines 147-153: You developed a new tool for assessment of Lifestyle, Coping Behavior, and Stressors Related. The description of the development process should be described in more details, including background and initial steps for psychometric properties establishment. At least data on face validity should be presented.

Response:

These items were developed in this study based on previous research (References 35–39), and we should examine the reliability and validity of these items in future studies. We added this problem as one of the study limitations (Lines 372–375). We were not able to investigate the validity of these items in our study; thus, we did not calculate the total score using these items but only used the score of each item for analysis.

Comment 9:

(Results) The groups were different as for the number of participants: 82%, 11%, 7% of the total sample size. Please describe statistical procedures that were applied through the analysis to address this difference in the groups’ size.

Response:

As per your comment, we should consider the difference in sample size between groups, and thus, we applied Scheffe’s method for multiple comparisons (Lines 193; 246–247; 256).

Comment 10:

(Results) Lines 175-176: You report on statistical significant differences in age between the groups. However, based on effect size magnitude, this difference has no practical significance. Please elaborate on this issue. 

Response:

As you pointed out, there is no significant difference in age between the groups, so this point was mentioned. (Lines 217–218)

Comment 11:

(Results) “Comparisons of psychological index scores between three AUDIT groups” – data which presented in the Table 3 – the statistical analysis is not sufficient to provide answer on the research question. More specifically, there are so many demographic differences between the groups, the analysis on simple difference between the groups does not allow to conclude on the source of the differences. For example, the between group differences in the depression scores may be attributed to the age differences rather than to the alcohol usage patterns. You should control through the analysis at least the main variables of age and gender.

Response:

Other reviewers also made similar comments about controlling for demographic data such as sex, and we conducted a multinomial logistic regression analysis based on your comments. (Lines 264–275, Table 6)

Comment 12:

(Results) “Comparisons of COVID-19 related variables between three AUDIT groups” – data presented in the table 4 – Again, the same issue of demographic between groups differences affected the results. Moreover, as it was described, the items of COVID-19 related variables were scored on the ordinal scale. Such type of the data does not fit the assumptions of one-way ANOVA analysis, which makes it even more difficult to conclude on something.

Response:

Thank you for your suggestion. However, many studies have treated Likert scale values as interval scales for convenience, and parametric tests have been performed on them (Carifio et al., 2008). (Strictly speaking, the total score of an existing scale is also just the sum of ordinal scales and not an interval scale.)

Carifio, J., & Perla, R. (2008). Resolving the 50‐year debate around using and misusing Likert scales. Medical education, 42(12), 1150-1152.

Thus, we adopted a general method in this study, which is in line with previous studies. We appreciate your understanding.

Comment 13:

(Results) Lines 263-265: “Moreover, depression and anxiety in women participants in this study were relatively higher than those in men, as well as when compared to the results found in many previous studies (Appendix A)” – If this information is so important to be discussed first, you should provide the data analysis in the manuscript body instead of appendix.

Response:

Based on your suggestion, we changed Appendix A to Table 3 and described the results. (Lines 221–223)

Comment 14:

(Results) Lines 274-288: the discussion on the between-groups differences in psychologic indices is less relevant until you re-do analysis with control for the between-groups differences in demographic data.

Response:

We conducted a multinomial logistic regression analysis and attempted to improve this problem.

Comment 15:

(Results) Lines 289: The differences in the COVID-19 measurements – since (a) the process of development of the tool is not evident through the manuscript and (b) the applied statistical approach seems to be less relevant, the results can be seen as unsupported by the data.

Response:

Based on your constructive comments, we tried to improve the manuscript as much as possible. Thank you for your useful advice.

Round 2

Reviewer 1 Report

The authors have greatly improved the manuscript. The considerations offered as a reviewer have been adequately addressed.

Author Response

Dear Reviewer 1,

Comment:

The authors have greatly improved the manuscript. The considerations offered as a reviewer have been adequately addressed.

Response:

Thank you for your constructive advice. Our paper is now of a higher quality. We are sincerely grateful to you.

Reviewer 2 Report

One minor change- authors can remove betas and SE from the multinomial logistic regression models given that they have already presented OR(95% CI).

Author Response

Dear Reviewer 2

Comment:

One minor change- authors can remove betas and SE from the multinomial logistic regression models, given that they have already presented OR(95% CI).

Response:

Thank you for your advice. We removed the betas and SE from Table 6.

Reviewer 3 Report

Thank the authors for addressing my comments.

The following changes are still needed:

Abstract

In the cross-sectional study, the term “prediction” can’t be used for the methodological issues. Please replace it.

Introduction

I appreciate the efforts the authors did to address my comments. However, still there is no description of the hazards of the alcohol use for the person and the community. In other words, why the addiction to the alcohol is an issue. In addition, please add information on patterns of alcohol use in Japan before the pandemic breakout.

Methods and Materials

Lifestyle, Coping Behavior, and Stressors scale: please add to the tool description statement that for the methodological of psychometric properties of the tool, each item was scored separately.

Statistical Analysis

“Multinomial logistic regression analyses with backward stepwise were conducted to examine the effects of hazardous a …” Please add “method” after the “stepwise”.

Results

Table 4 add important information. As it presented, low effect size was found for all statistical between-groups differences in the study. Please elaborate on this issue. The same is for the data that presented in the table 5.

Logistic regression report should be revised and the term “predict” should be replaced. In my opinion, logistic regression did not add new information, but MANCOVA analysis for AUDIT groups with intercept of gender, age groups, type of occupation, level of income, and marital status should be done. Otherwise, the conclusion is not rigorous.

Discussion

The Discussion presents the findings modestly based on the limitations.

Author Response

Dear Reviewer 3

We appreciate the constructive comments provided. We have revised the manuscript based on your suggestions, and the revised text is indicated in red.

Comment 1:

[Abstract] In the cross-sectional study, the term “prediction” can’t be used for the methodological issues. Please replace it.

Response:

As you advised, we replaced “prediction” to “related to” or “associated with”. (Line 25)

Comment 2:

[Introduction] I appreciate the efforts the authors did to address my comments. However, still there is no description of the hazards of the alcohol use for the person and the community. In other words, why the addiction to the alcohol is an issue. In addition, please add information on patterns of alcohol use in Japan before the pandemic breakout.

Response:

As you suggested, we have described risks posed by problems with alcohol use. (Line 61-66) Additionally, we have added information on alcohol use in Japan before the pandemic. (Line 77-82)

Comment 3:

[Methods and Materials] Lifestyle, Coping Behavior, and Stressors scale: please add to the tool description statement that for the methodological of psychometric properties of the tool, each item was scored separately.

Response:

As advised, we have added an explanation for these items. (Line 195-196)

Comment 4:

[Statistical Analysis] “Multinomial logistic regression analyses with backward stepwise were conducted to examine the effects of hazardous a …” Please add “method” after the “stepwise”.

Response:

As you advised, we added the word “method” (Line 213).

Comment 5:

[Results] Table 4 add important information. As it presented, low effect size was found for all statistical between-groups differences in the study. Please elaborate on this issue. The same is for the data that presented in the table 5.

Response:

As you pointed out, the effect sizes shown in Tables 4 and 5 are generally low. This may be due in part to the strong influence of demographic characteristics on the dependent variable, which increased the variance of the values in each group. Analysis adjusting demographic characteristics may be essential to explore the relationship between alcohol use and psychological indexes or items related to lifestyle, coping behavior, and stressors related to mild lockdown conditions (i.e., multinomial regression analysis or MANCOVA, which we conducted in this study). We added this explanation in the “Discussion” section. (Line 382-390)

Comment 6:

[Results] Logistic regression report should be revised and the term “predict” should be replaced. In my opinion, logistic regression did not add new information, but MANCOVA analysis for AUDIT groups with intercept of gender, age groups, type of occupation, level of income, and marital status should be done. Otherwise, the conclusion is not rigorous.

Response:

As you advised, we replaced “predict” with “relate” (Line 325, 335, 352, 363, 367, 375, and 414). We conducted MANCOVA, but the Box’s M value of 1475.14 was associated with a p-value of <0.001, which indicated that the covariance matrices between the groups were not assumed to be equal for the MANCOVA. Therefore, we have added a description of this result. (Line 209-212 and 280-284)

Comment 7:

[Discussion] The Discussion presents the findings modestly based on the limitations.

Response:

Given the limitations of this study, it is necessary to avoid categorical statements in the discussion section, so we made changes to present our findings using a more modest tone. (Line 342, 343, and 372)